# Orally Dissolving Film-Based Influenza Vaccines Confer Superior Protection Compared to the Oral Administration of Inactivated Influenza Virus

**DOI:** 10.3390/vaccines13060600

**Published:** 2025-05-31

**Authors:** Keon-Woong Yoon, Jie Mao, Gi-Deok Eom, Su In Heo, Ki Back Chu, Mi Suk Lee, Fu-Shi Quan

**Affiliations:** 1Department of Biomedical Science, Graduate School, Kyung Hee University, Seoul 02447, Republic of Korea; keon-woong.yoon@yale.edu (K.-W.Y.); maojie@khu.ac.kr (J.M.); ekd3910@khu.ac.kr (G.-D.E.); hsi7200@khu.ac.kr (S.I.H.); 2Department of Microbial Pathogenesis, Yale University School of Medicine, New Haven, CT 06510, USA; 3Medical Research Center for Bioreaction to Reactive Oxygen Species and Biomedical Science Institute, Core Research Institute (CRI), Kyung Hee University, Seoul 02447, Republic of Korea; 4Department of Parasitology, College of Medicine, Inje University, Busan 47392, Republic of Korea; kbchu@inje.ac.kr; 5Department of Infectious Disease and Malaria, Paik Institute of Clinical Research, Inje University, Busan 47392, Republic of Korea; 6Division of Infectious Diseases, Department of Internal Medicine, College of Medicine, Kyung Hee University, Seoul 02447, Republic of Korea; mslee7@khu.ac.kr; 7Department of Medical Zoology, School of Medicine, Kyung Hee University, Seoul 02447, Republic of Korea

**Keywords:** orally dissolving film, vaccine, influenza virus, mucosal immunity

## Abstract

**Background**: Self-administered orally dissolving films (ODFs) encapsulating inactivated influenza vaccines represent an effective strategy for stimulating mucosal immunity. While this vaccination method offers several advantages over conventional influenza vaccines, a comparative efficacy study remains lacking. **Methods**: Female BALB/c mice were immunized with inactivated A/PR/8/34 (H1N1) either via orogastric inoculation or through the oral mucosal delivery using pullulan and trehalose-based ODF vaccines. Each group received equivalent antigen doses across three immunizations. Humoral responses and antibody functionality were assessed using sera collected post-immunization. After lethal viral challenge, other immunological and virological parameters were determined in corresponding tissues. Body weight and survival were monitored over a 14-day period after challenge. **Results**: ODF vaccination elicited significantly higher virus-specific IgA levels, HAI titers, and neutralizing antibody activity than oral gavage. After the viral challenge, ODF-immunized mice exhibited stronger IgG and IgA responses in respiratory tissues, increased antibody-secreting cells in lungs and spleen, and elevated germinal center B cells and CD8^+^ T cell responses. Both vaccination methods reduced lung pro-inflammatory cytokines and provided full protection against lethal challenge; however, the ODF group showed lower cytokine levels, better weight maintenance, and reduced viral loads. **Conclusions**: ODF vaccination elicits more robust systemic and mucosal immune responses than oral vaccination and may serve as a promising alternative method of influenza vaccine delivery.

## 1. Introduction

Influenza virus continues to pose a significant global health threat and is one of the leading causes of seasonal respiratory infections. Vaccination remains the primary intervention strategy to mitigate influenza-related morbidity, hospitalization, and mortality, particularly among the elderly and individuals with chronic diseases [1]. Despite strong recommendations from the World Health Organization for annual influenza vaccination of high-risk groups, global coverage rates remain suboptimal, especially in low- and middle-income countries [2,3]. A major barrier to broader vaccine uptake is the reliance on conventional intramuscular delivery route, which is invasive, often poorly accepted by recipients, and requires trained personnel and disposable medical supplies. These challenges are further compounded in resource-limited settings, where shortages of personnel, infrastructure, and logistical capacity constrain large-scale immunization efforts [4].

To address these limitations, needle-free vaccination strategies are actively being explored. Among them, oral immunization through the gastrointestinal tract offers a non-invasive, user-friendly alternative that reduces logistical complexity and healthcare expenditure [5]. Preclinical studies have demonstrated that oral administration of inactivated influenza antigen can induce both systemic immune responses and mucosal immune responses. However, antigen degradation by gastric acid, digestive enzymes, and bile salts, substantially reduces bioavailability, thereby limiting immunogenicity [5,6]. This inherent limitation underscores the need for novel delivery strategies that can preserve antigen structure or bypass gastrointestinal transit altogether.

Orally dissolving films (ODFs) have emerged as a promising alternative to conventional oral delivery system, offering both structural advantages and immunological advantages over the conventional influenza vaccines. These solid-state formulations, composed of biocompatible film-forming agents, effectively maintain the structural integrity and bioactivity of embedded substances by minimizing degradation and enhancing stability under variable environmental conditions [7]. ODFs dissolve rapidly upon contact with saliva, eliminating the need for water intake and thereby enhancing user convenience and improving compliance, especially in individuals with swallowing difficulties [8]. Their flexible film format also enables easy dose customization and simplifies storage and transportation [9].

Beyond these structural advantages, ODFs enable direct antigen delivery to the oral mucosa, bypassing the harsh gastrointestinal environment and hepatic first-pass metabolism. This localized delivery enhances antigen stability and exposure at mucosal inductive sites, which may lead to improved immunogenicity [9]. In our previous work, we incorporated inactivated influenza antigens into a pullulan–trehalose-based ODF matrix. This formulation elicited robust immune responses and conferred complete protection against lethal influenza virus challenge in animal models, with preliminary evidence of a dose-sparing effect [10,11]. However, a direct comparison between ODF vaccination and conventional oral administration has not yet been conducted. Building on these prior findings, the present study employed a murine model to compare the immunogenicity and protective efficacy of ODF-based vaccines versus orogastric vaccine delivery. This investigation aimed to further evaluate the translational potential of ODFs as a practical, recipient-friendly alternative for influenza vaccine administration.

## 2. Materials and Methods

### 2.1. Ethics Statement

Animal experimental schedule was conducted in strict accordance with the Guide for the Care and Use of Laboratory Animals (National Institutes of Health) and was approved by the Institutional Animal Care and Use Committee of Kyung Hee University (accession number: KHUASP-SE-18-024). Mice were housed in a specific pathogen-free facility with a controlled environment (22 ± 2 °C, 50 ± 10% humidity, and 12 h light/dark cycle) and were given ad libitum access to chow and water. All efforts were made to minimize suffering. Prior to all invasive procedures, mice were anesthetized using inhaled isoflurane. Mice exhibiting signs of severe distress, reaching a humane intervention point defined as a loss of more than 25% of initial body weight, or those at the end of the study, were humanely euthanized via CO_2_ inhalation followed by cervical dislocation.

### 2.2. Cell Maintenance and Virus Preparation

Madin-Darby Canine Kidney (MDCK) cells (ATCC, Manassas, VA, USA) were cultured in Dulbecco’s Modified Eagle Medium (DMEM; Welgene, Gyeongsan, South Korea) supplemented with 10% fetal bovine serum (FBS), 100 U/mL penicillin, and 100 μg/mL streptomycin (p/s) and maintained at 37 °C in a humidified incubator with 5% CO_2_. Influenza A/Puerto Rico/8/1934 (H1N1, PR8) virus was propagated in an 11-day-old SPF embryonated chicken eggs as previously described [11]. Diluted virus stock was injected into the allantoic cavity and the eggs were incubated at 37 °C for 2.5 days. Following incubation, the eggs were chilled at 4 °C for 12 h. Allantoic fluid was then collected and clarified by centrifugation at 4750 rpm for 30 min at 4 °C. The harvested supernatants were either used directly for animal challenge experiments or subjected to formalin inactivation. Inactivated PR8 virus particles were purified using a 60%/30%/15% sucrose density gradient and were subsequently used as coating or stimulating antigens in immunological assays, as well as a component of the ODF vaccines.

### 2.3. Formulation of ODF Vaccines

ODF vaccines were prepared as previously described with minor optimizations [10,11]. Briefly, pullulan (Tokyo Chemical Industry, Tokyo, Japan) and trehalose (Sigma-Aldrich, St. Louis, MO, USA) were dissolved in distilled water to form the film-forming matrix. Each formulation contained 12 μg of pullulan and 3 μg of trehalose per 20 μL of solution. Inactivated PR8 virus was added to the solution at a final dose of 100 μg to generate a homogeneous liquid vaccine formulation. For control films, 100 μg of bovine serum albumin (BSA) was incorporated instead of PR8. The vaccine solution was cast onto a parafilm surface in 3 mm × 3 mm squares and air-dried at room temperature for 4 h in a Class II biosafety laminar flow cabinet. The resulting solid ODF vaccines were detached, characterized by measurements of thickness, weight, and HA titer before use in immunization experiments.

### 2.4. Hemagglutination (HA) Assay

The functional integrity of the antigen incorporated into the ODF vaccines was evaluated using a standard HA assay [11]. Antigen samples, including the inactivated PR8 virus in liquid formulation and reconstituted ODF vaccines, were diluted 1:100 in phosphate-buffered saline (PBS). A volume of 50 μL aliquot of each diluted sample was added to the first column of a U-bottom 96-well microtiter plate, followed by serial two-fold dilutions across the plate. Then, 50 μL of the 0.5% chicken RBC (cRBC) suspension was added to each well, and plate was incubated at room temperature for 1 h. HA titers were determined as the highest dilution factor at which complete hemagglutination was observed. Titers were expressed as log_2_ values. Negative controls included PBS and blank film formulations containing only pullulan and trehalose. An equivalent amount of inactivated PR8 virus without any ODF components was used as the positive control.

### 2.5. Immunization and Challenge

Thirty-two female BALB/c mice (SPF, 7 weeks old, 18–20 g) were purchased from Nara Biotech (Seoul, South Korea). Each individual mouse was considered an experimental unit. Following one week of acclimation, mice were randomly assigned to two experimental groups (ODFV and Oral; n = 8 per group) and two control groups (Naive and Naive + Challenge; n = 8 per group). The total sample size was determined based on prior literature involving preclinical vaccine studies [10,11]. Experimental groups were immunized with inactivated influenza PR8 virus, either via oral gavage (diluted in 100 μL PBS) or using the ODF platform. For ODF administration, a single vaccine film was placed on the tongue of mice and held in place until complete dissolution [10,11]. All groups received three immunizations at four-week intervals, with each dose containing an equivalent antigen amount (50 μg). Naive group received PBS alone, while the Naive + Challenge group served as an unimmunized infection control. Peripheral blood was collected one week after each immunization to evaluate systemic antibody responses, and an additional blood sample was obtained two weeks after the final dose to assess hemagglutination inhibition (HAI) and virus-neutralizing activities. At four weeks after the final immunization, all groups except the naive control were intranasally challenged with 5 × LD_50_ of live PR8 virus in a 50 μL volume. At 4 days post-infection (dpi), four mice from each group were randomly selected and euthanized for the collection of tissues used in immunological and virological analyses. The remaining mice were monitored for 14 days post-challenge to assess daily body weight changes and survival outcomes.

All mice were housed and handled under standardized conditions to minimize environmental variability. Immunizations, viral challenges, and sample collections were conducted simultaneously across all groups, with treatments administered in a random order to reduce procedural bias. While cage location and handling order were not systematically varied, all procedures were performed according to consistent protocols to limit potential confounding factors. No priori inclusion or exclusion criteria were defined. All animals enrolled in the study completed the full immunization and challenge schedule, and no experimental units or data points were excluded from the analysis. All collected samples met quality control standards and were included in the final dataset.

### 2.6. Sample Collection and Processing

Blood samples were collected via retro-orbital plexus puncture after each immunization. Samples were centrifuged at 6000 rpm for 10 min to isolate serum, which was used to assess antibody responses and functional analyses. At 4 dpi, mice were euthanized, and mucosal samples were collected by flushing the trachea. Lung tissues were homogenized in PBS, filtered through a 100 μm cell strainer, and centrifuged at 2000 rpm for 10 min. Supernatants were collected for subsequent pro-inflammatory cytokine and viral burden analyses. Cell pellets were washed and resuspended in diluted Percoll for density gradient centrifugation to isolate lung lymphocytes. Cell pellets from spleens were collected using the same procedure and treated with RBC lysis buffer. Final single cell suspensions from lung and spleen samples were used for antibody-secreting cell (ASC) assays and flow cytometric analysis of immune cell populations. Group allocation was known to the investigators throughout sample collection and subsequent analyses, and blinding was not applied.

### 2.7. Virus-Specific Antibody Response Measurement

PR8 virus-specific IgG and IgA antibody responses in serum and mucosal samples were quantified using an enzyme-linked immunosorbent assay (ELISA) [12]. Briefly, 96-well immunoplates were coated with 5 μg/mL of inactivated PR8 virus diluted in carbonate-bicarbonate coating buffer (pH 9.6). After coating, plates were washed three times with PBS containing 0.05% Tween-20 (PBST) and blocked with 0.2% gelatin in PBST for 2 h at 37 °C. Samples were serially diluted in PBS and added to the plates, followed by incubation at 37 °C for 1 h. After washing with PBST, bound antibodies were detected using horseradish peroxidase (HRP)-conjugated goat anti-mouse IgG or IgA in 1:2000 dilution for 1 h incubation at 37 °C. Color development was achieved using citrate–phosphate buffer containing o-phenylenediamine (OPD and H_2_O_2_). Optical density was measured at 450 nm using a microplate reader (Biochrom Ltd., Cambridge, UK).

### 2.8. HAI and Virus Neutralization Assays

To evaluate functional antibody responses against influenza virus, HAI and virus neutralization assay were conducted as previously described [10]. Briefly, sera were treated with receptor-destroying enzyme (Denka Seiken, Tokyo, Japan) according to the manufacturer’s instructions. For HAI assay, treated sera was serially diluted and mixed with 4 HAU of PR8 virus suspension, followed by incubation for 30 min at room temperature. Following incubation, 0.5% cRBCs were added to each well and incubated for an additional hour. The HAI titer was defined as the reciprocal of the highest serum dilution that completely inhibited HA. Virus neutralization was determined using a plaque reduction assay. Serially diluted sera were mixed with 100 plaque-forming units (PFU) of live PR8 virus and incubated at 37 °C for 1 h to allow neutralization. The serum-virus mixtures were then added to confluent MDCK cell monolayers and adsorbed for 1 h at 37 °C with 5% CO_2_. After removing the infection inoculum, cells were overlaid with medium supplemented with 2× DMEM, DEAE-dextran, MEM NEAA, p/s, glutamine, and trypsin. Plates were incubated at 37 °C for 4 days, after which cells were fixed with 4% paraformaldehyde for 30 min and stained with 1% crystal violet to visualize plaques. The number of plaques in each well was counted, and the percentage of neutralization was calculated relative to the plaque number in wells without serum treatment.

### 2.9. ASC Response Evaluation

Single-cell suspensions from lungs and spleens were counted using hemocytometers and seeded at a density of 1 × 10^6^ cells per well into 96-well flat-bottom plates pre-coated with inactivated PR8 virus antigen (5 μg/mL). Cells were cultured at 37 °C in a 5% CO_2_ incubator for 5 days. To detect PR8-specific ASCs, cell culture supernatants were removed and incubated with HRP-conjugated anti-mouse IgG (1:1000 dilution) for 1 h at 37 °C. After washing with PBST, OPD substrate solution was added to each well for color development. Absorbance was measured at 450 nm using a microplate reader (Biochrom Ltd., Cambridge, UK) [10].

### 2.10. Flow Cytometry Analysis of Immune Cell Populations

Single-cell suspensions from lungs and spleens were prepared as previously described and adjusted to a concentration of 1 × 10^6^ cells per 100 μL in RPMI-1640 medium (Welgene) supplemented with 10% FBS and 1% p/s [10]. To enhance surface marker expression and antigen-specific activation, cells were stimulated with inactivated PR8 virus (5 μg/mL) for 2 h at 37 °C before FcR blocking. After 15 min of blocking, cells were stained for 30 min at 4 °C in PBS buffer containing 2% FBS and 0.1% sodium azide with fluorochrome-conjugated monoclonal antibodies purchased from BD Biosciences (San Jose, CA, USA): CD3-FITC, CD4-PE-Cy7, CD8a-PE, B220-FITC, GL7-PE, CD45-FITC, CD19-PE-Cy7, IgD-PE, and CD38-Alexa647. Flow cytometric data were acquired using an Accuri C6 flow cytometer and analyzed with C6 Accuri software (version 227.4, BD Biosciences). Lymphocytes were gated based on forward and side scatter profiles, and data were reported as the percentage of each stained population within the total lymphocytes. To calculate absolute cell numbers, the total lymphocyte count per sample was multiplied by the corresponding percentage of each gated subset.

### 2.11. Pro-Inflammatory Cytokine Measurement

Lung tissue homogenates were prepared from mice at 4 dpi. Concentrations of pro-inflammatory cytokines IFN-γ and IL-6 in the lung supernatants were quantified using commercial BD OptEIA™ ELISA kits (BD Biosciences), following the manufacturer’s instructions. Cytokine concentrations were calculated based on standard curves generated from recombinant cytokine standards provided with the kits.

### 2.12. Lung Viral Titer Determination

Viral burdens were determined by plaque assay using the supernatants from lungs collected at 4 dpi [10]. Briefly, MDCK cells were seeded in 12-well tissue culture plates and grown to 90–100% confluence. Prior to infection, cells were washed with PBS and inoculated with 10-fold serial dilutions of lung homogenates prepared in DMEM medium. Virus adsorption was carried out for 1 h with gentle rocking every 15 min. Following adsorption, the inoculum was removed, and cells were overlaid with semisolid medium. Plates were incubated for 3–4 days at 37 °C in a 5% CO_2_ incubator, fixed with 4% paraformaldehyde for 30 min, and stained with 1% crystal violet solution for plaque visualization.

### 2.13. Statistical Analysis

All data were analyzed using GraphPad Prism software (version 9.0, GraphPad Software, San Diego, CA, USA). Continuous variables were expressed as mean ± SD. Data normality was assessed using the Shapiro–Wilk test, and all datasets were confirmed to follow a normal distribution prior to further analysis. The significance between the groups was determined using either Student’s *t*-test or a one-way analysis of variance with Bonferroni’s post hoc test. Representative data from 4 individual animal experiments were provided. Statistical significance was defined as * *p* < 0.05, ** *p* < 0.01, and *** *p* < 0.001.

## 3. Results

### 3.1. Characterization of ODF Vaccines

A pure concentrate of inactivated influenza PR8 virus (V) was formulated with pullulan (P) and trehalose (T) in distilled water, and 100 μL aliquots of the resulting mixture (Liquid + P + T + V) were cast onto PET plastic films. Following 4 h of drying, a solidified carbohydrate matrix with an approximate size of 3 × 3 mm was obtained (ODFV), incorporating 100 μg of inactivated PR8 virus per unit (Figure 1A). Control formulations included films containing only P and T (P + T), and P + T supplemented with 50 μg BSA (P + T + BSA), to assess the impact of antigen loading on the physical and biological characteristics of the films. Despite the presence or absence of protein antigens, no notable differences in film thickness were observed among the three formulations (Figure 1B). However, the addition of either BSA or PR8 virus, led to a significant increase in film weight compared to P + T alone, suggesting successful embedding of protein content (Figure 1C). No significant weight difference was detected between BSA-loaded and virus-loaded ODFs.

The functional activity of the incorporated antigen was evaluated via HA titer assay. Neither the P + T formulation nor PBS exhibited any HA activity, confirming the absence of inherent agglutinating components in the film base. In contrast, the inactivated PR8 virus retained robust HA activity with a titer of 15 in the liquid ODF, which was comparable to the virus alone without excipients. After drying, the HA titer of the resulting ODF slightly decreased to 14 (Figure 1D). Although a minor loss of HA activity was observed during the drying process, the antigen remained functionally active within the solid matrix. These results demonstrated that the ODF platform successfully incorporated and stabilized viral antigen within a dissolvable carbohydrate matrix while maintaining structural uniformity and preserving antigenic functionality.

### 3.2. ODF Vaccination Elicited Higher Humoral and Functional Immune Responses than Oral Delivery

To compare the immunogenicity and protective efficacy of different vaccination routes, female BALB/c mice were immunized with inactivated PR8 influenza virus via oral gavage or ODF. Each group was immunized three immunizations at four-week intervals with equal antigen doses. Sera were collected after each immunization to assess virus-specific antibody titers and functional antibody responses, including HAI and viral neutralization. Four weeks after the final immunization, mice were intranasally challenged with 5 × LD_50_ of live PR8 virus. At 4 dpi, lung, spleen, and mucosal tissues of mice were harvested for virological and immunological analyses. The remaining animals were monitored daily until 14 dpi to assess survival and body weight changes as indicators of protective efficacy (Figure 2A). Both vaccination strategies significantly elevated serum IgG levels compared to the naive control group, with peak titers observed after the third dose. No significant difference was found between Oral and ODFV groups (Figure 2B). Notably, virus-specific IgA responses were detectable after the first dose in the ODF group, whereas no such increase was observed in the Oral group at this stage. After booster immunizations, both groups exhibited significantly increased IgA responses, with the ODFV group consistently exhibiting significantly higher IgA levels than the Oral group (Figure 2C). A similar trend was observed in HAI activity. While both vaccination routes increased HAI titers with subsequent doses, the ODFV group exhibited significantly higher titers from the first boost onward, which were sustained through the final dose (Figure 2D). Virus neutralization assays further confirmed the superior functional antibody responses induced by ODFV. At 1:10 serum dilution, complete plaque inhibition was observed in both groups. However, at 1:30 dilution, sera from the ODFV group maintained 100% neutralization efficacy, whereas virus neutralization dropped to 70% in the Oral group. At 1:90 dilution and beyond, neutralization capacity decreased in both groups. However, ODFV immune sera retained around 40% plaque reduction at a 1:270 dilution, while sera from the Oral group dropped below 20% (Figure 2E). These results demonstrate that ODF vaccination induces stronger humoral immune responses and superior functional antibody activity compared to conventional oral delivery.

### 3.3. ODF Vaccination Induced Superior Mucosal Antibody Responses than Oral Delivery

To assess mucosal immune responses elicited by different immunization routes, respiratory tract samples were collected at 4 dpi with a lethal dose of PR8 virus. In the trachea, ODFV induced higher levels of IgG and IgA compared to both the naive challenge control and Oral groups. In contrast, oral vaccination led to a significant increase in IgA levels relative to the naive challenge control but failed to induce detectable IgG responses (Figure 3A,B). Similarly, in lung supernatants, robust virus-specific IgG and IgA responses were observed in the ODFV group, while the Oral group displayed negligible antibody levels (Figure 3C,D). Collectively, these results illustrate that ODFV effectively induces strong mucosal antibody responses in the respiratory tract, with immunogenicity superior to oral delivery.

### 3.4. ODF Vaccination Stimulated Stronger ASC Responses in the Lungs and Spleens than Oral Delivery

To determine the capacity of different vaccination routes to elicit functional B cell responses, ASC activity was evaluated in murine lungs and spleens following immunization and subsequent viral challenge. Single-cell suspensions were isolated from the lung and spleen tissues and cultured in vitro with influenza antigen stimulation to quantify virus-specific IgG and IgA-secreting plasma cells. In the lungs, the ODFV group induced significantly elevated ASC responses for both IgG and IgA compared to the naive challenge control, confirming its ability to stimulate local plasma cell differentiation. In contrast, oral vaccination failed to induce detectable ASC responses in the lungs (Figure 4A,B). A similar trend was observed in the spleens, where ODFV immunization elicited markedly higher IgG and IgA ASC responses than either oral delivery or unimmunized control. Although the oral group showed a modest increase in ASC response, the differences were not statistically significant relative to the controls (Figure 4C,D). These results indicate that ODFV induces functional plasma cell responses in both mucosal and systemic compartments, highlighting its immunological advantage over oral vaccination.

### 3.5. ODF Vaccination Enhanced GC B Cell and CD8^+^ T Cell Responses in the Lungs than Oral Delivery

To evaluate GC-associated and cell-mediated immune responses induced by distinct vaccination routes, flow cytometry was performed to assess the frequencies of immune cell subsets in lung tissues at 4 dpi. Both vaccination strategies significantly elevated the frequencies of GC B cells and CD4^+^ T cells relative to the Naive + Challenge group in lung tissues, suggesting a general enhancement of local adaptive immunity (Figure 5A–D). Notably, ODF vaccines induced a significantly higher GC B cell proportion than oral vaccination (Figure 5B), whereas both routes activated helper T cells to comparable levels (Figure 5D). Importantly, only the ODFV demonstrated a significant elevation in CD8^+^ T cell frequencies in the lungs relative to the control, while oral vaccination failed to induce such an elevation (Figure 5E). These findings highlight that ODFV platform is more effective in engaging cellular immunity at the site of infection than oral administration.

### 3.6. ODF Vaccination More Effectively Suppressed Pulmonary Inflammation and Conferred Superior Protection in Infected Mice than Oral Delivery

Following lethal influenza virus challenge (4 dpi), pro-inflammatory cytokines, including IFN-γ and IL-6, were markedly elevated in lung homogenates of unvaccinated mice, whereas only baseline levels were detected in the naive control (Figure 6A,B). Both immunization routes significantly reduced inflammatory cytokine levels relative to the naive challenge control, and IFN-γ levels were comparable between the two immunization groups (Figure 6A). However, IL-6 levels were significantly higher in the Oral group than those in ODFV group (Figure 6B). Consistent with IL-6 concentrations, lung viral titers were substantially reduced in both vaccinated groups compared to the unimmunized control (Figure 6C). Of the two immunization groups, ODFV mice exhibited significantly lower viral loads than the Oral mice. Body weight changes were monitored daily after challenge as a marker of disease severity (Figure 6D). The ODFV route maintained body weights of immunized mice within the normal range throughout the observation period. In contrast, mice in the naive challenge group exhibited progressive weight loss beginning at 3 dpi, reaching the humane endpoint by 8 dpi. While oral vaccination provided partial protection, mice in this group began losing weight at 6 dpi, with the most pronounced decline at 7 dpi. Nevertheless, weight loss in this group did not exceed 10% of initial body weight and fully recovered by 10 dpi, at which point no significant difference in body weight loss was observed between the two vaccinated groups. All immunized mice survived the 14-day monitoring period, resulting in a 100% survival rate (Figure 6E). In comparison, 50% of unvaccinated mice succumbed to infection by 7 dpi, and the remaining animals either died or reached humane endpoints by 8 dpi. Taken together, these findings demonstrate that ODF vaccination more effectively limits IL-6-mediated inflammation, suppresses viral replication, and mitigates disease manifestations including weight loss than the oral vaccination route.

## 4. Discussion

Our present study systematically compared the immunogenicity and protective efficacy of a previously developed ODF vaccine with conventional oral administration route in a murine model. The ODF vaccine platform elicited significantly stronger systemic and mucosal immune responses, enhanced viral control, and more effective protection against lethal influenza challenge, supporting its feasibility and immunological advantages as a mucosal influenza vaccine.

The ODF formulation used in this study was developed through iterative optimization of earlier designs consisting of pullulan and trehalose, which allow precise antigen dosing and needle-free administration without requiring trained personnel. Our previous prototypes employed relatively large (0.8–1 cm^2^) and thick (up to 350 μm) film matrices [10,11]. In contrast, the current formulation features a smaller (3 mm × 3 mm), thinner (~110 μm) design with a reduced casting volume of 20 μL. This optimized format enhances ease of administration, reduces storage and transportation demands, and may enhance mucosal adherence and antigen dispersion through increased surface contact during dissolution [13,14].

Immunologically, ODF vaccines induced earlier and more robust virus-specific IgA responses than oral delivery. IgA was detectable after a single ODF and increased with boosters, whereas oral immunization required multiple doses to elicit modest IgA responses, which remained significantly lower. This suggests that antigen delivery via the oral mucosa achieved by ODF promotes early mucosal-immune activation. This advantage may be attributed to the controlled dissolution of the solid-state matrix, enabling prolonged antigen exposure and enhancing immune cell activation [15,16]. Furthermore, direct absorption through the oral mucosa not only accelerates antigen uptake but also helps preserve antigen integrity by bypassing exposure to the harsh gastrointestinal conditions [17,18]. These features likely facilitate early IgA class switching and robust boosting of mucosal antibody responses.

Importantly, ODF vaccines elicited stronger mucosal immunity at the primary site of influenza infection, as evidenced by elevated virus-specific IgG and IgA levels in the respiratory tract. Secretory antibodies play a key role in neutralizing viruses and preventing epithelial attachment, thereby limiting viral replication [19,20]. This enhanced response may result from fundamental differences in antigen delivery between the two routes. Oral vaccination exposes antigens to gastric degradation, impairing immune recognition and reduces efficacy [21,22]. In contrast, ODF vaccines deliver intact antigens directly to the oral mucosa, thereby enhancing antigen stability, facilitating local presentation, and promoting drainage to cervical lymph nodes [23].

These cervical nodes, though not anatomically linked to the respiratory tract, are functionally associated with respiratory immunity and may facilitate immune cell migration via mucosal cross-talk with nasal-associated lymphoid tissue [24,25,26]. Consistent with this, ODF immunization resulted in significantly higher ASC and GC B cell activity in the lungs, indicating enhanced B cell activation and affinity maturation [27]. Contrastingly, oral vaccination failed to induce comparable ASC response despite similar lung CD4^+^ T cell frequencies, suggesting inefficient T–B cell interaction or a skewing of CD4^+^ T cell subsets to gut-associated lymphoid tissues [28,29].

Cell-mediated immunity was also more effectively stimulated by ODF vaccines. Notably, only ODF vaccination induced a significant increase in CD8^+^ T cells. Given the critical role of lung tissue-resident CD8^+^ T cells in mediating rapid viral clearance and recovery [30,31], their induction likely also contributed to the lower pulmonary inflammation, reduced viral loads, and diminished tissue damage observed in the ODF group.

Despite the encouraging immunological outcomes, further investigations are warranted to fully establish the potential of the ODF platform. Evaluating its protective breadth against heterologous and heterosubtypic influenza strains will better reflect real-world applicability. In addition, long-term immune monitoring with dose sparing effects is essential to fully assess durability and practical utility of ODF-based vaccines.

## 5. Conclusions

In conclusion, ODF vaccines elicited stronger humoral and cellular immune responses and conferred superior protective efficacy against lethal influenza virus challenge compared to conventional oral administration, highlighting their promise as an alternative mucosal influenza vaccine platform.

## Figures and Tables

**Figure 1 vaccines-13-00600-f001:**
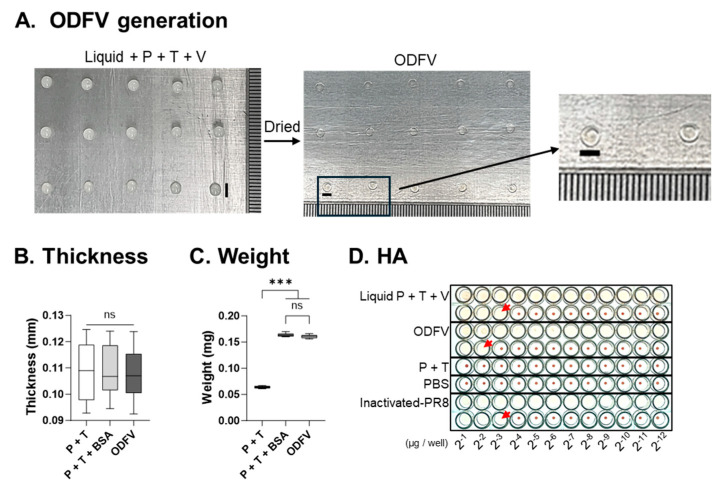
Characterization of orally dissolving film vaccine (ODFV). (**A**) Preparation of ODFV by casting a solution containing inactivated A/PR/8/34 virus, pullulan (P), and trehalose (T) onto PET film (Liquid + P + T +V), followed by drying (ODFV). (**B**) Thickness and (**C**) weight of ODFV were determined. Formulations containing only T and P, or supplemented with an equivalent amount of BSA, were used as controls. (**D**) Hemagglutination (HA) assay was used to compare HA activity before and after drying, as well as against control groups. Red arrows indicate wells in which hemagglutination took place. Statistical analysis was performed using one-way analysis of variance with Bonferroni’s post hoc test (GraphPad Prism 9.0). Data represent mean ± SD. ns, no significance; *** *p* < 0.001.

**Figure 2 vaccines-13-00600-f002:**
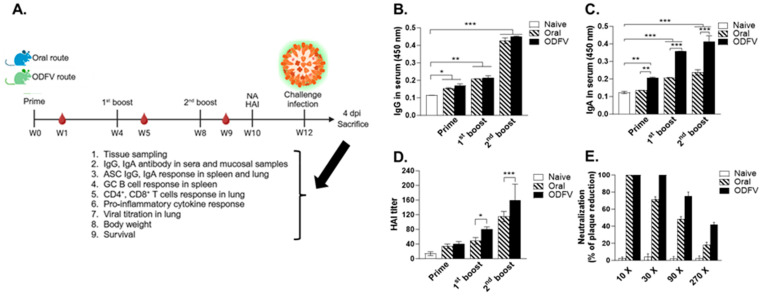
Systemic antibody responses and functional antibody activity following immunization. (**A**) Experimental schedule. Mice were immunized three times with inactivated PR8 virus via orally dissolving films (ODFV route) or oral gavage (Oral route) at 4-week intervals. Four weeks after the last dose, all mice were intranasally challenged with a lethal dose of live virus. At 4 days post-infection (dpi), mice were euthanized for collection of spleen and lung tissues to evaluate subsequent immune responses and protective efficacy. Sera were collected one week after each immunization to evaluate virus-specific IgG (**B**) and IgA (**C**) levels using ELISA. Sera were also obtained two weeks after the third immunization to measure hemagglutination inhibition (HAI) titers (**D**) using a standard HAI assay, and virus neutralization activity (**E**) was assessed by plaque reduction assay, presented as the percentage reduction in plaque formation. For between-group comparisons at each immunization time point, and one-way ANOVA with Bonferroni’s post hoc test for within-group comparisons across time points. For each analysis, n = 8 per group, with each sample measured in triplicate. Data represent mean ± SD. * *p* < 0.05; ** *p* < 0.01; *** *p* < 0.001.

**Figure 3 vaccines-13-00600-f003:**
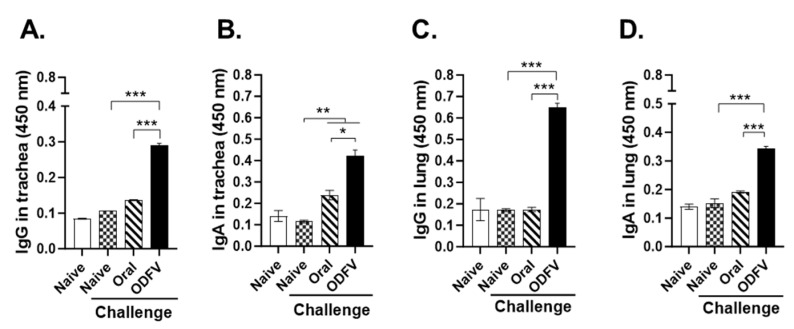
Mucosal antibody responses in the respiratory tract. Mice were euthanized at 4 days post-challenge to collect trachea and lung tissues. Virus-specific IgG and IgA levels in (**A**,**B**) tracheal washes and (**C**,**D**) lung supernatants were quantified by ELISA. Statistical analysis was performed using one-way analysis of variance with Bonferroni’s post hoc test (GraphPad Prism 9.0). For each analysis, n = 4 per group, with each sample measured in triplicate. Data represent mean ± SD. * *p* < 0.05; ** *p* < 0.01; *** *p* < 0.001.

**Figure 4 vaccines-13-00600-f004:**
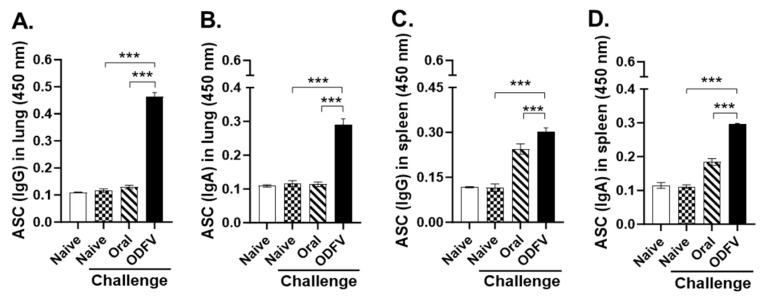
Antigen-specific antibody-secreting cell (ASC) responses. At 4 days post-challenge, single-cell suspensions were isolated from lungs and spleens and restimulated in vitro with inactivated PR8 virus. After incubation for 3–5 days, secreting IgG and IgA in (**A**,**B**) lungs and (**C**,**D**) spleens were measured using cell culture supernatant. Statistical analysis was performed using one-way analysis of variance with Bonferroni’s post hoc test (GraphPad Prism 9.0). For each analysis, n = 4 per group, each sample was measured in triplicate. Data represent mean ± SD. *** *p* < 0.001.

**Figure 5 vaccines-13-00600-f005:**
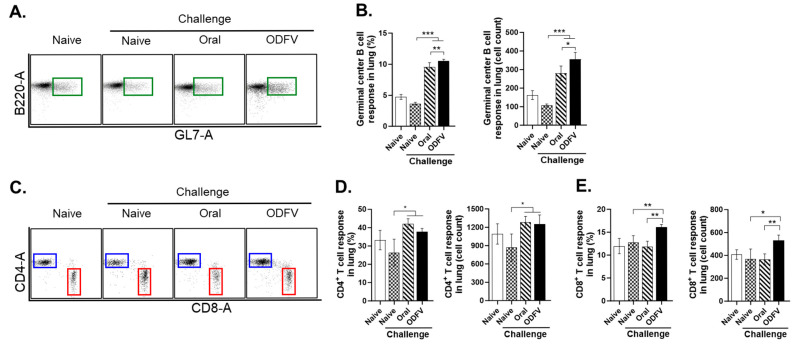
Germinal center B cell and T cell responses. Lung single-cell suspension was prepared at 4 days post-challenge to evaluate adaptive immune responses. (**A**,**B**) Germinal center B cell (B220^+^GL7^+^), (**C**,**D**) CD4^+^ T cell (CD3^+^CD4^+^), and (**C**,**E**) CD8^+^ T cell (CD3^+^CD8^+^) proportions in lung lymphocytes were identified and quantified by flow cytometry, with both their proportions and absolute cell numbers determined. Lung lymphocytes were gated based on forward and side scatter profiles. The green, blue, and red squares represent the gating regions for identifying the corresponding positive cell populations. Statistical analysis was performed using one-way analysis of variance with Bonferroni’s post hoc test (GraphPad Prism 9.0). For each analysis, n = 4 per group, each sample was measured in triplicate. Data represent mean ± SD. * *p* < 0.05; ** *p* < 0.01; *** *p* < 0.001.

**Figure 6 vaccines-13-00600-f006:**
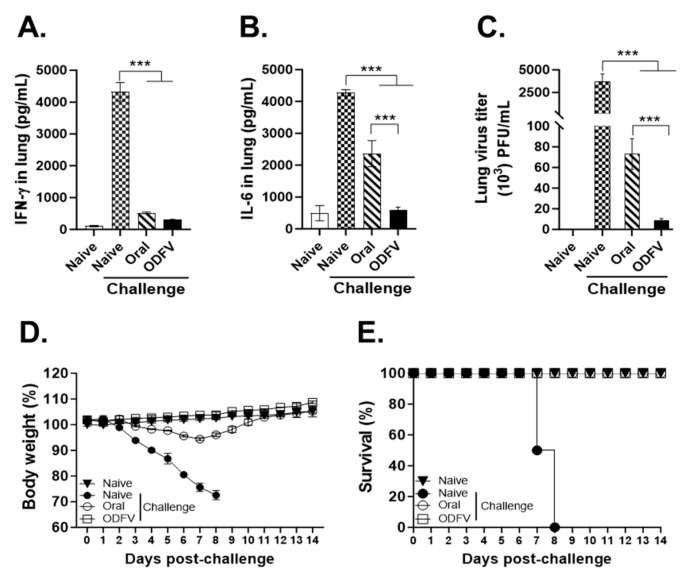
Protective efficacy conferred by distinct routes of vaccination. Lung tissues were collected at 4 days post-challenge to assess inflammatory status and viral replication. Pro-inflammatory cytokine (**A**), IFN-γ, and (**B**) IL-6 concentrations in lung homogenates were measured by cytokine ELISA. (**C**) Viral titters in lung tissues were determined by plaque assay. (**D**) Body weight reduction and (**E**) survival were recorded daily for 14 days after infection to monitor disease progression and evaluate vaccine efficacy. Statistical analysis was performed using one-way analysis of variance with Bonferroni’s post hoc test (GraphPad Prism 9.0). For each analysis, n = 4 per group, each sample was measured in triplicate. Data represent mean ± SD. *** *p* < 0.001.

## Data Availability

Data will be made available on request.

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
