# Peer review of "Orally Dissolving Film-Based Influenza Vaccines Confer Superior Protection Compared to the Oral Administration of Inactivated Influenza Virus"

_vaccines, 2025, doi:10.3390/vaccines13060600_

Round 1

Reviewer 1 Report

Comments and Suggestions for Authors

Yoon et al compared ODF vaccination and conventional oral administration of inactivated influenza virus vaccine by studying IgG, IgA, HAI after immunization and IgG, IgA and CD4+ T cells and GC B cells as well as protection after challenge. They found ODF vaccination can elicit stronger humoral and cellular immunity and provide better protection against viral challenge compared with conventional oral administration. Overall, the manuscript was well-written. I only have some minor questions.

  1. Figure 2, the ELISA reading values look quite low. Is there any positive control to confirm the assay? Or is the IgG titer considered low?
  2. Please provide full name of Naïve+Cha for readability.
  3. Line 354, is it mononuclear cells or just single cell suspension? How the mononuclear cells were isolated?
  4. Figure 5A and 5C, what is the parent population for the percentage?
  5. Figure 5D, is there a significant difference between Naive and Naive+Cha group? How do you explain the decreased CD4+ T cells (compared with naive) in naive+cha group?

Author Response

1.Figure 2, the ELISA reading values look quite low. Is there any positive control to confirm the assay? Or is the IgG titer considered low?

Response Although the absolute OD values of virus-specific IgG may appear relatively low, these levels can be influenced by various factors including coating antigen concentration, serum dilution, incubation time, and assay sensitivity. Importantly, our focus is on vaccine efficacy, where the key metric is the fold increase in antibody titers compared to the unvaccinated group, rather than the absolute titer values.

In our previous studies using the same coating concentration (5 μg/mL of inactivated virus), an approximately 3–5-fold increase in IgG level relative to the naïve control has been previously reported in similar studies and is considered potentially associated with protection against lethal influenza challenge (International Journal of Pharmaceutics, 2024, 667: 124945 Fig. 2A ~3-fold increase with a lower vaccination dose. Antiviral Research, 2024, 230: 105979, Fig. 4A, ~4-fold increase). Similarly, in this study, the IgG levels in vaccinated mice were approximately four times higher than in naive controls, which falls within the expected protective range. Therefore, a separate positive control was not set in the present study.

Moreover, antibody functionality was further confirmed by HAI and virus neutralization assays (Figures 2D and 2E), both indirectly supporting the validity of the ELISA data. We hope this addresses the reviewer’s concern and confirms the reliability of our assay results.

2.Please provide full name of Naïve+Cha for readability.

Response: As requested by the reviewer, we have revised all "Naive+Cha" in the manuscript and figures to "Naive+Challenge" to ensure clarity and consistency (lines 340,342,386; figure 3-6).

3.Line 354, is it mononuclear cells or just single cell suspension? How the mononuclear cells were isolated?

Response: We apologize for the incorrect wording and have corrected 'mononuclear cells' to 'single-cell suspensions' (lines 360, 375, 395). No mononuclear cell isolation was performed in this study.

4.Figure 5A and 5C, what is the parent population for the percentage?

Response: The percentages shown in Figure 5A and 5C are based on the gated lymphocyte population. We have clarified this in revised “Methods” and figure legend accordingly (lines 228-230, 398-400).

5.Figure 5D, is there a significant difference between Naive and Naive+Cha group? How do you explain the decreased CD4+ T cells (compared with naive) in naive+cha group?

Response: Although the CD4 T cell frequency in the Naive+Challenge group appears lower than in the Naive group, statistical analysis indicated that this difference is not significant (one-way analysis of variance with multiple Bonferroni’s post hoc test). The visual decrease is probably due to natural biological variability within a relatively small sample size (n = 4).

Reviewer 2 Report

Comments and Suggestions for Authors

The paper describes an improved method of oral administration of influenza vaccines. Orally dissolving films (ODFs) was elaborated, encapsulating inactivated influenza vaccines. ODFs have emerged as a promising alternative to conventional oral delivery system, offering both structural advantages and immunological advantages over the conventional influenza vaccines. ODFs enable direct antigen delivery to the oral mucosa, bypassing the harsh gastrointestinal environment. Antigen delivery via the oral mucosa achieved by ODF promotes early mucosal immune activation. ODF vaccines elicited stronger mucosal immunity at the primary site of influenza infection, as evidenced by elevated virus-specific IgG and IgA levels in the respiratory tract. Cell-mediated immunity was also more effectively stimulated by ODF vaccines. Solid-state formulations, effectively maintain the structural integrity and bioactivity of embedded substances by minimizing degradation and enhancing stability under variable environmental conditions. A direct comparison between ODF vaccination and conventional oral administration was performed.

Note

Line 191   Optical density was measured at 450 nm using a microplate reader  

Why was optical density measured at 450 nm if o-phenylenediamine was used, whose color reaction is usually measured at 492 nm?

Author Response

Line 191   Optical density was measured at 450 nm using a microplate reader 

Why was optical density measured at 450 nm if o-phenylenediamine was used, whose color reaction is usually measured at 492 nm?

Response: Although 492 nm represents the peak absorbance of OPD reaction products, 450 nm also falls within the effective absorbance range and provides sufficient sensitivity for accurate quantification. This wavelength was selected based on the configuration of our microplate reader, which is calibrated for high-sensitivity detection at 450 nm, and consistency with prior validated protocols (Antiviral Research, 2024, 230: 105979. Fig.4; Emerging Microbes & Infections, 2025: 2494702. Fig. 2B). Nonetheless, we acknowledge the reviewer’s comments and would optimize the detection wavelength in our further studies.

Reviewer 3 Report

Comments and Suggestions for Authors

The manuscript reported an orally dissolving film (ODF)-based influenza vaccine using the inactivated H1N1 virus that induced substantially higher virus-specific IgA responses as well as virus neutralizing activity and provided better protection against lethal challenge compared to the oral administration virus in mice.  It would provide an alternative strategy for influenza vaccine oral delivery.  There are some major concerns to be answered for publication.

  1. The authors prepared the ODF vaccine by casting a solution containing the inactivated PR8 virus, pullulan, and trehalose onto PET film followed by drying, which was characterized by determining its thickness, weight as well as HA titer (Fig. 1). How the ODF vaccine could be mass-produced as well as its quality control.  Furthermore, it is suggested that the authors provide more data for the vaccine characterization, such as virus loading ratio and ODF stability etc.
  2. The authors used the inactivated PR8 virus diluted in PBS as an oral gavage control (Line 143). The vaccine without any protectant is easily destroyed by gastric acid, digestive enzymes and bile salts thereby limiting immunogenicity as the authors mentioned (Lines 58-59). So it is better to add applicable oral vaccine protectant in the oral gavage control to compare the real different effect induced by the two administration strategies.
  3. After the lethal homologous challenge, the lung viral burden as well as body weight and survival were used to evaluate protection effect induced by the influenza vaccinations. However, the authors need compare the pathological changes in the lungs and trachea to provide solid evidence for the protection.

Author Response

The authors prepared the ODF vaccine by casting a solution containing the inactivated PR8 virus, pullulan, and trehalose onto PET film followed by drying, which was characterized by determining its thickness, weight as well as HA titer (Fig. 1). How the ODF vaccine could be mass-produced as well as its quality control. Furthermore, it is suggested that the authors provide more data for the vaccine characterization, such as virus loading ratio and ODF stability etc.

Response: Viral loading ratio reviewer suggested has been provided in detail in the manuscript (lines 117–118 and 265). Data for HA titer in Figure 1D. and protection in Figure 6 indicated that ODF were stable to induce a vaccine efficacy. ODF stability has been also investigated in our previous studies (Antiviral Research, 2024, 230: 105979. Fig. 3; International Journal of Pharmaceutics, 2024, 124945) which have been cited (line 114). The mass-production on ODF vaccine and its quality control will be studied soon. 

The authors used the inactivated PR8 virus diluted in PBS as an oral gavage control (Line 143). The vaccine without any protectant is easily destroyed by gastric acid, digestive enzymes and bile salts thereby limiting immunogenicity as the authors mentioned (Lines 58-59). So it is better to add applicable oral vaccine protectant in the oral gavage control to compare the real different effect induced by the two administration strategies.

Response: We fully agree that incorporating a protectant could improve the immunogenicity of orally delivered vaccines. However, the purpose of our study was not to compare different delivery matrices, but rather to directly assess the immunological outcomes of two delivery routes (oral gavage versus oral mucosal delivery via ODF) using the same antigen dose without additional formulation variables. This approach allowed us to clearly demonstrate the immunological advantages of ODF-based delivery over traditional oral administration under equivalent antigenic conditions. Although the comparisons mentioned by the reviewer are beyond the scope of this current work, we would like to add protective formulations for oral gavage delivery in further studies.

After the lethal homologous challenge, the lung viral burden as well as body weight and survival were used to evaluate protection effect induced by the influenza vaccinations. However, the authors need compare the pathological changes in the lungs and trachea to provide solid evidence for the protection.

Response: We evaluated pulmonary inflammation and pathology indirectly by quantifying pro-inflammatory cytokines (IFN-γ and IL-6) in lungs (lines 403-409, Figure 6A–B). These cytokines are well-established biomarkers of tissue inflammation and damage following influenza virus infection (Viruses, 2021, 13(7): 1362; Veterinary microbiology, 2000, 74(1-2): 109-116). We appreciate the reviewer’s suggestion and will incorporating histopathological analysis in future studies to further strengthen the evidence.

Reviewer 4 Report

Comments and Suggestions for Authors

Vaccines-3650061

The present manuscript compares the efficacy of an oral Flu vaccine in a formulation of orally dissolving film (ODF) with the orally administered Flu vaccine that involves the gastric mucosa. The model is well applied by vaccinating the mice via the tongue or gavage. The data are clear and convincing. However, some aspects need to be addressed. Specific comments are as follows:

  1. Line 141. The authors need to indicate what “+Cha” means for the reader.
  2. Line 152. The word “Naïve” should be written as naïve throughout the text.
  3. Line 272. The viral titer of 215 HAU is unclear. Viral titers are expressed in Log scale (Log2 or Log10). Please justify the titer representation or change to the conventional expression.
  4. Figure 2A. The protocol should indicate if the “oral route” refers to oral gavage. Otherwise, it is not very clear.
  5. Figures 3-6 would be more precise if the groups second-naïve, Oral, and ODFV are underlined with a +Challenge. Even if in figure 2A, the “oral and ODFV” are indicated as challenged. Having only naïve+Cha is confusing.
  6. Figure 3B. For consistency, the bar of ODFV should be the same color as the rest of the graphs in this figure.
  7. Line 384. The authors mention that “…oral vaccination failed to elicit an effective cytotoxic T cell response.” This assumes that all CD8 T cells are cytotoxic, which is not the case. Moreover, no experiments have tested the cytotoxic capability of the CD8 T cells. So, the differences are only in the frequency of the CD8 T cells, not in their activity. That should be corrected.
  8. Figure 5, panels B, D, E. The authors should include the calculated total number of each population, in addition to the already included percentages.
  9. Figure 5. The labels of this figure are off since they go from A to C, then to B, D, and E. It may help the reader if the graphs are in sequential order.

Author Response

The present manuscript compares the efficacy of an oral Flu vaccine in a formulation of orally dissolving film (ODF) with the orally administered Flu vaccine that involves the gastric mucosa. The model is well applied by vaccinating the mice via the tongue or gavage. The data are clear and convincing. However, some aspects need to be addressed. Specific comments are as follows:

1.Line 141. The authors need to indicate what “+Cha” means for the reader.

Response: To avoid confusion, we have revised the abbreviation 'Cha' with 'Challenge' throughout the manuscript (lines 340, 342, 386) and figures.

2.Line 152. The word “Naïve” should be written as naïve throughout the text.

Response: As suggested, we have revised “Naïve” to its lowercase version. Moreover, we replaced “naïve” with “naive” throughout the manuscript for consistency with the simplified American English spelling (lines 304, 408).

However, we did not modify cases where the term appears at the beginning of a sentence or is used as part of a specific group label. If the reviewer still recommends changing these instances, we’d like to revise accordingly.

3.Line 272. The viral titer of 215 HAU is unclear. Viral titers are expressed in Log scale (Log2 or Log10). Please justify the titer representation or change to the conventional expression.

Response: HA titers are now reported as log values (lines 275 and 277), with the definition provided in the Methods section (lines 129-131).

4.Figure 2A. The protocol should indicate if the “oral route” refers to oral gavage. Otherwise, it is not very clear.

Response: We have newly indicated that ‘Oral route refers to oral gavage in Figure legend of 2A according to the suggestion (line 326).

5.Figures 3-6 would be more precise if the groups second-naïve, Oral, and ODFV are underlined with a +Challenge. Even if in figure 2A, the “oral and ODFV” are indicated as challenged. Having only naïve+Cha is confusing.

Response: We have revised Figure 3-6 according to the suggestion.

6.Figure 3B. For consistency, the bar of ODFV should be the same color as the rest of the graphs in this figure.

Response: We have revised Figure 3B according to the suggestion.

7.Line 384. The authors mention that “…oral vaccination failed to elicit an effective cytotoxic T cell response.” This assumes that all CD8 T cells are cytotoxic, which is not the case. Moreover, no experiments have tested the cytotoxic capability of the CD8 T cells. So, the differences are only in the frequency of the CD8 T cells, not in their activity. That should be corrected.

Response: We have revised the description related to CD8 T cells in lines 391–392 to ensure accurate interpretation of the results.

8.Figure 5, panels B, D, E. The authors should include the calculated total number of each population, in addition to the already included percentages.

Response: In addition to percentages, we have newly calculated and added the absolute cell numbers for immune cells to the revised figure (Figure 5) and corresponding legends and methods (lines 230-232, 399).

9.Figure 5. The labels of this figure are off since they go from A to C, then to B, D, and E. It may help the reader if the graphs are in sequential order.

Response: The current panel sequence (A–E) was intentionally arranged to group each flow cytometry plot with its corresponding quantification, such that Panels A–B represent the germinal center B cell data, and Panels C–E display the T cell analyses.

We believe this structure maintains logical consistency by visually linking related data.

Round 2

Reviewer 3 Report

Comments and Suggestions for Authors

No comments.